# Phylogenomics of 10,575 genomes reveals evolutionary proximity between domains Bacteria and Archaea

Qiyun Zhu [1,19], Uyen Mai[2,19], Wayne Pfeiffer[3], Stefan Janssen [1,4], Francesco Asnicar [5], Jon G. Sanders [1], Pedro Belda-Ferre [1], Gabriel A. Al-Ghalith [6], Evguenia Kopylova[1], Daniel McDonald[1], Tomasz Kosciolek [1,7], John B. Yin[8,9], Shi Huang[1,10], Nimaichand Salam[11], Jian-Yu Jiao[11], Zijun Wu[1,12], Zhenjiang Z. Xu[1], Kalen Cantrell[6], Yimeng Yang [6], Erfan Sayyari[8], Maryam Rabiee[2], James T. Morton[1,2], Sheila Podell [13], Dan Knights[6], Wen-Jun Li [11], Curtis Huttenhower [14,15], Nicola Segata [5], Larry Smarr[2,16,17], Siavash Mirarab [7] & Rob Knight [1,2,16,18]*

Rapid growth of genome data provides opportunities for updating microbial evolutionary relationships, but this is challenged by the discordant evolution of individual genes. Here we build a reference phylogeny of 10,575 evenly-sampled bacterial and archaeal genomes, based on a comprehensive set of 381 markers, using multiple strategies. Our trees indicate remarkably closer evolutionary proximity between Archaea and Bacteria than previous estimates that were limited to fewer "core" genes, such as the ribosomal proteins. The robustness of the results was tested with respect to several variables, including taxon and site sampling, amino acid substitution heterogeneity and saturation, non-vertical evolution, and the impact of exclusion of candidate phyla radiation (CPR) taxa. Our results provide an updated view of domain-level relationships.

[1] Department of Pediatrics, University of California San Diego, La Jolla, CA, USA. [2] Department of Computer Science and Engineering, University of California San Diego, La Jolla, CA, USA. [3] San Diego Supercomputer Center, University of California San Diego, La Jolla, CA, USA. [4] Algorithmic Bioinformatics, Department of Biology and Chemistry, Justus Liebig University Gießen, Giessen, Germany. [5] Department CIBIO, University of Trento, Trento, Italy. [6] Department of Computer Science and Engineering, University of Minnesota, Minneapolis, MN, USA. [7] Malopolska Centre of Biotechnology, Jagiellonian University, Krakow, Poland. [8] Department of Electrical and Computer Engineering, University of California San Diego, La Jolla, CA, USA. [9] Department of Mathematics, University of California San Diego, La Jolla, CA, USA. [10] Single-Cell Center, Qingdao Institute of Bioenergy and Bioprocess Technology, Chinese Academy of Sciences, Qingdao, Shandong, China. [11] State Key Laboratory of Biocontrol and Guangdong Key Laboratory of Plant Resources, School of Life Sciences, Sun Yat-sen University, Guangzhou, China. [12] Division of Biological Sciences, University of California San Diego, La Jolla, CA, USA. [13] Scripps Institution of Oceanography, University of California San Diego, La Jolla, CA, USA. [14] Department of Biostatistics, Harvard T. H. Chan School of Public Health, Boston, MA, USA. [15] The Broad Institute of MIT and Harvard, Cambridge, MA, USA. [16] Center for Microbiome Innovation, University of California San Diego, La Jolla, CA, USA. [17] California Institute for Telecommunications and Information Technology, University of California San Diego, La Jolla, CA, USA. [18] Department of Bioengineering, University of California San Diego, La Jolla, CA, USA. [19]These authors contributed equally: Qiyun Zhu, Uyen Mai. *email: robknight@ucsd.edu

The metaphor of a "tree of life" was used by Darwin in his *On the Origin of Species* in 1859. It came into its modern form when Carl Woese and co-workers used the new ability to genetically sequence the small subunit (SSU) ribosomal RNA gene from multiple different organisms to create a phylogenetic tree[1], thereby showing a scenario of three domains of life: Bacteria, Archaea, and Eukaryota[2,3]. Recent years have seen discoveries of novel microbial groups enabled by culture-based and metagenomic methods[4–7], many of which represent previously unknown biodiversity[4,8,9], and keep updating our knowledge of the extent and relationships among domains as indicated by phylogenetics[10–13]. Among these new discoveries is the candidate phyla radiation (CPR, also referred to as Patescibacteria)[4,8], a highly diversified clade of mainly uncultivated microorganisms that may subdivide the domain of Bacteria[11], although this scenario remains controversial[14]. Meanwhile, the discovery and analysis of multiple novel archaeal lineages have suggested an archaeal origin for eukaryotes, pointing to a two-domain scenario[15,16]. The currently representative view of the tree of life, inferred based on the concatenation of ribosomal proteins, illustrated a bipartite pattern with distinct separation between Bacteria (including CPR) and Archaea (plus Eukaryota)[11,13]. More comprehensive work in both taxon and locus sampling exists[14], but the inter-domain relationships were not explored.

Reconstructing phylogenies typically relies on comparing homologous features. Although closely related organisms often share obvious genome-level homologies, building higher-level, especially cross-domain phylogenies has been challenging due to the rarity of clearly defined homologies[17]. To date, many efforts rely on one, or a few, universal "core" genes that are usually involved in fundamental translation machinery[15,18]. Examples include the SSU rRNA[19–21] and several dozens ribosomal proteins[22]. The choice of these "marker genes" is based on their universality, conservation, and the observation that they suffer from less frequent horizontal gene transfer (HGT)[17]. However, HGT is widespread across the domains[19–21], affecting even the most conserved "housekeeping" genes[23], and cannot be ruled out even for these markers. Furthermore, the reliance on a few marker genes limits the information (i.e., phylogenetic signal) available for resolving all relationships in the tree of life. Finally, it reduces applicability in metagenomics—increasingly the main source of novel genome data—where assembled genomes are frequently incomplete and error-prone[6]. Maximizing the included number of loci, thus, is desirable. However, when dealing with many loci, reconciling discordant evolutionary histories among different parts of the genome can become challenging. Moreover, a practical dilemma is imposed by computational limitations: adding breadth across the phylogenetic space requires more computing effort, which leads to compromises with either the quantity of genes analyzed[11] or the robustness of tree-building algorithms[14].

In this work, we build a reference phylogeny of 10,575 bacterial and archaeal genomes (Fig. 1). They are sampled from all 86,200 nonredundant genomes available from NCBI GenBank and RefSeq[24] as of March 7, 2017 (Fig. 2), using a statistical approach that maximizes the covered biodiversity. Our phylogenetic reconstruction uses 381 marker genes, selected from whole genomes solely by sufficient sequence conservation to identify homology. The whole data set totals 1.16 trillion non-gap amino acids, making it among the largest single data sets upon which de novo phylogenetic trees have been built (Supplementary Table 1). To infer species trees, we use both a summary approach that accounts for discrepancy among the evolutionary histories of individual genes, and the conventional gene alignment concatenation approach. The resulting species trees provide high resolution of the basal relationships among microbial clades,

which show that Bacteria and Archaea are in closer proximity compared with previous estimations (Fig. 1). The phylogeny also enable us to evaluate and revise previously established taxonomic hierarchies. We have made our data and protocols publicly available at https://biocore.github.io/wol/.

## Results

**Comprehensive sampling of biodiversity and genes**. By using a purpose-built "prototype selection" algorithm to maximize evenness of genome sampling (Supplementary Fig. 1, detailed in Supplementary Note 1) and by incorporating multiple additional criteria, including marker gene presence, genome quality, and taxonomy, we selected 10,575 genomes, covering 146 of 153 phyla defined by NCBI, plus all 89 classes, 196 of 199 orders, 422 of 429 families, 2081 of 2117 genera, and 9105 of 20,779 species (Fig. 2a). A total of 2852 genomes (27.0%) are metagenome-assembled genomes (MAGs), while the remaining are from isolates and other sources (Fig. 2c). Meanwhile, 2267 genomes (21.4%) are complete genomes or chromosomes, while the remaining are scaffolds or contigs (Fig. 2d). Overall, the selected genomes are of high completeness and low contamination as evaluated based on known lineage-specific marker gene sets (Fig. 2b). By testing against the MAG quality standard established by Bowers et al.[6], only 10.4% MAGs or 3.7% of all genomes fall within the low-quality draft category, while the remaining meet the criteria of either high- or medium-quality drafts (Fig. 2e). This balanced representation of known bacterial and archaeal diversity ensured the comprehensiveness and evenness of the resulting phylogeny.

Our phylogenomic analysis was based on the 400 marker genes originally proposed in PhyloPhlAn[25] (Supplementary Fig. 2). The taxon sampling protocol ensured that all selected genomes contain at least 100 marker genes each. In the resulting data matrix, each marker gene is present in $7565 \pm 1730$ (mean and std. dev.) genomes (Supplementary Fig. 2a), while each genome contains $286.14 \pm 80.23$ (mean and std. dev.) marker genes (Supplementary Fig. 2b). These marker genes were further filtered down to 381, based on metrics of alignment quality (see the Methods section) across the sampled genomes (Supplementary Fig. 2d).

**Assessing deep phylogeny using multiple strategies**. We explored multiple tree inference methods (detailed in Supplementary Note 2, with selected ones compared in Fig. 3 and Supplementary Fig. 3), but will mostly focus on two strategies: CONCAT and ASTRAL. CONCAT concatenates gene alignments and infers a single tree using maximum likelihood (ML) performed using the robust implementation in RAxML[26]. Computational limitations forced us (Supplementary Table 2) to use at most 100 sites per gene, selected either randomly ("concat.rand") or based on maximum conservation ("concat.cons"). However, we also tested analyzing all sites, using the faster but less accurate ML program, FastTree[27] (referred to as "fasttree"). In contrast, the ASTRAL tree ("astral") is based on first inferring 381 gene trees and then summarizing them using the ASTRAL method[28]. ASTRAL accounts for gene tree discordance due to divergent coalescent histories and has been shown in simulations to be more accurate than concatenation in the presence of highly frequent HGTs[29]. Due to its inherent scalability, ASTRAL analyses were able to use all the data (i.e., all sites of every gene). For comparison with previous studies[11], a CONCAT tree was also built using 30 ribosomal proteins ("concat.rpls"). We used ML to estimate branch lengths for the ASTRAL tree based on the same data used to infer the CONCAT tree.

Overall, ASTRAL (Fig. 1; Supplementary Fig. 4) and CONCAT trees (Supplementary Figs. 5, 6) show congruence in topology

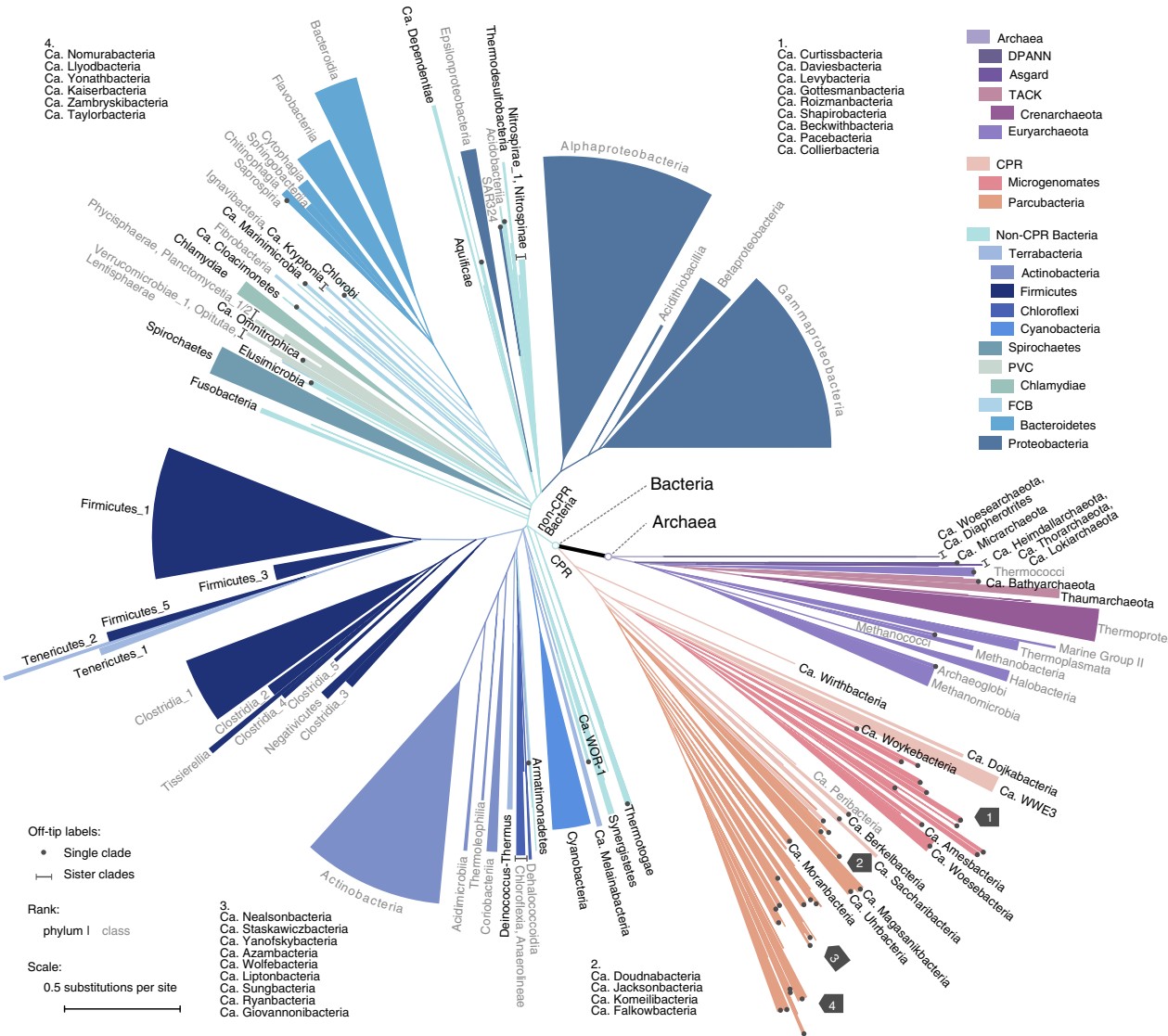

**Fig. 1** A new view of the bacterial and archaeal tree of life. The tree contains 10,575 evenly distributed bacterial and archaeal genomes, with topology reconstructed using ASTRAL based on individual trees of 381 globally sampled marker genes, and branch lengths estimated based on 100 most conserved sites per gene. Branches with effective number of genes (en) ≤ 5 and local posterior probability (lpp) ≤ 0.5 were collapsed into polytomies. Taxonomic labels at internal nodes and tips reflect the tax2tree curation result. Color codes were assigned to above-phylum groups and phyla with 100 or more representatives. To display the tree in a page, it was collapsed to clades (sectors) representing phyla with at least one taxon (black), and classes with at least ten taxa (gray). The radius of a sector indicates the median distance to all descending taxa of the clade, and the angle is proportional to the number of descendants. For polyphyletic taxonomic groups, minor clades with <5% descendants of that of the most specious clade were omitted, while the remaining clades were appended a numeric suffix sorted by the number of descendants from high to low. Dots (single clade) and lines (sister clades) are used to assist visual connection between tips and labels, where necessary. In four instances where visual space is inadequate (marked by gray arrows), groups of labels in clockwise order are provided in remote blank areas. Source data are provided as a Source Data file.

(Fig. 3a, b; Supplementary Note 2) when compared with trees derived from implicit (e.g., distance-based) analyses (Supplementary Fig. 7, Supplementary Note 2). The congruence is higher at shallow branches, but generally decreases as phylogenetic depth increases (Supplementary Fig. 8). The ASTRAL tree, in particular, has high support among the early branching clades (Supplementary Figs. 3, 9, also see Supplementary Figs. 4–6). This high resolution is directly related to the large number of gene trees used in the inference, as using fewer loci notably decreased the branch support of the species tree (Supplementary Fig. 10, Supplementary Note 2). On the other hand, the evolutionary relationships recovered by CONCAT are impacted by the breadth of site sampling (Supplementary Fig. 11, Supplementary Note 2)

and the robustness of method (Supplementary Fig. 12, Supplementary Note 2).

To further evaluate the impact of taxon sampling, we tested a series of subsampled sets of taxa, selected so that they maximize the representation of large and deep-branching clades (see the Methods section). Reducing taxon sampling changed the overall topology (Supplementary Fig. 13, Supplementary Note 2) and the inferred relationship between large groups (e.g., placement of Chloroflexi and Chlamydiae) (Supplementary Fig. 14), further highlighting the importance of our dense sampling of genomes.

Phylogenetic trees built by both strategies recapitulated clear separation between Archaea (669 taxa) and Bacteria (9906 taxa) at the root (Figs. 1, 3, Supplementary Figs. 4–6). Meanwhile, CPR

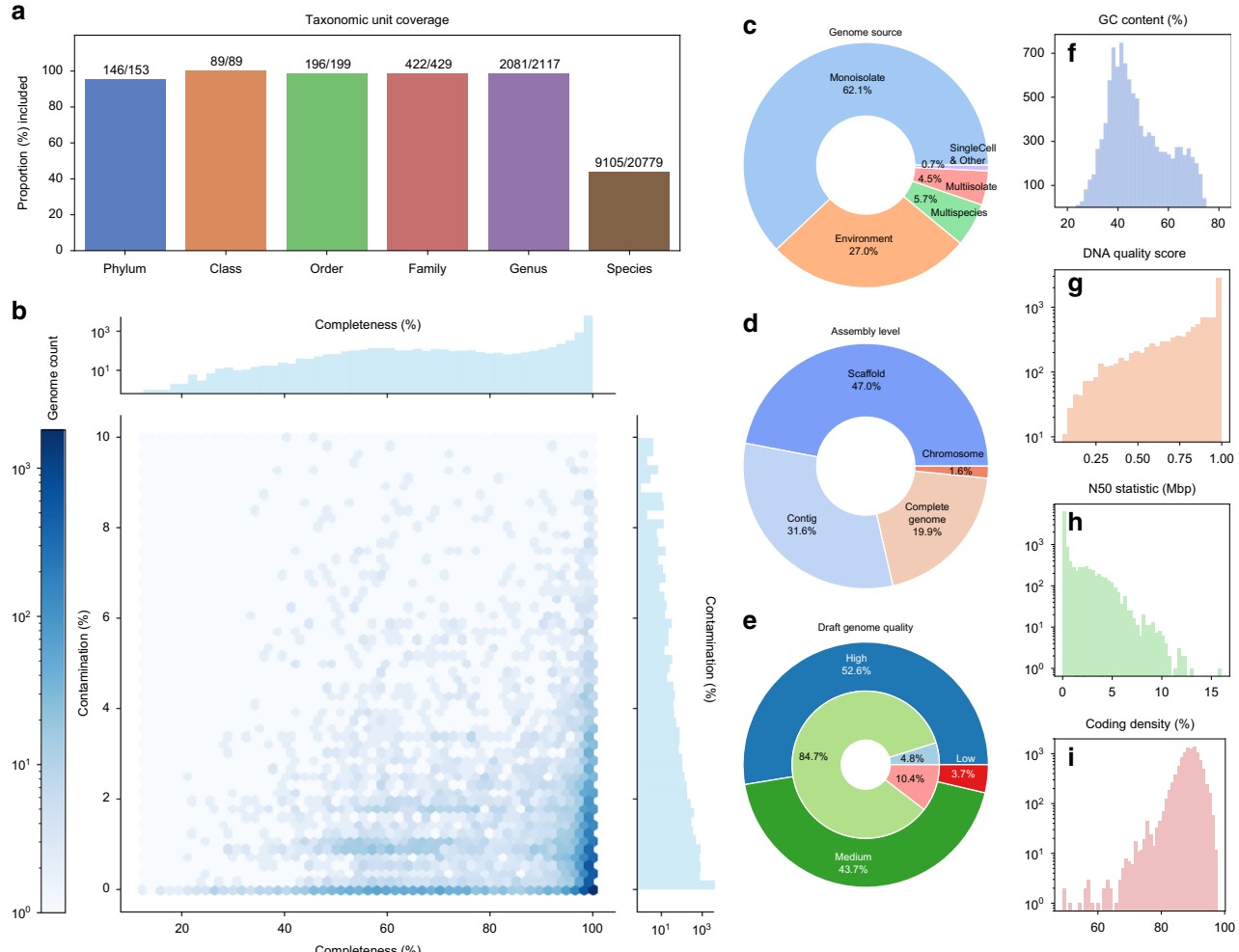

**Fig. 2** Statistics of the 10,575 bacterial and archaeal genomes selected for phylogenetic reconstruction. **a** Numbers and proportions of NCBI-defined taxonomic units included from all 86,200 available genomes. **b** Distribution of completeness vs. contamination scores computed by CheckM[44]. **c** Distribution of genome sources, i.e., the "scope" property defined by NCBI. **d** Distribution of genome assembly levels. **e** Distribution of draft genome quality, determined following the standard established by Bowers et al.[6]. Specifically: "high": completeness > 90%, contamination < 5%, presence of 23S, 16S, 5S rRNAs, and ≥18 tRNAs; "medium": completeness ≥ 50%, contamination < 10%; "low": completeness < 50%, contamination < 10%. The outer, darker circle represents all genomes. The inner, lighter circle represents genomes assembled from metagenomes (MAGs). **f** Distribution of GC contents. **g** Distribution of DNA quality scores, calculated following Land et al.[41]. **h** Distribution of N50 statistics of nucleotide sequences per genome. **i** Distribution of coding density. The y-axes in **f**–**h** represent genome counts. Source data are provided as a Source Data file.

(1454 taxa) forms a monophyletic group located at the base of the bacterial lineage in the ASTRAL tree and the CONCAT trees that use global markers (Figs. 3c, 4a). Considering the potential impact of long-branch attraction, this placement will require further validation using more robust substitution models and controlled tests. The ASTRAL tree shows high consistency and moderate-to-high branch support for several taxonomic units recently defined above the phylum level, including TACK, Microgenomates, Parcubacteria, FCB, PVC, and Terrabacteria[4] (Fig. 3c, d). These groups were also supported in the CONCAT trees, with the exception of Terrabacteria in one analysis (Fig. 3c, d). With reference to the trees, we systematically evaluated and curated NCBI taxonomy, showing frequent incongruences (Supplementary Fig. 15a, c, Supplementary Table 3), especially in metagenome-derived genomes (detailed in Supplementary Note 3). We further compared our trees with the recently developed GTDB taxonomy and trees[14], and observed overall high congruence, though with a few exceptions at deep branches (Fig. 3a–d; Supplementary Figs. 15b, d, 16, elaborated in Supplementary Note 4). A detailed interpretation of our

phylogeny in reference to taxonomy and multiple previous works is provided in Supplementary Note 5.

**Evolutionary proximity between Archaea and Bacteria.** ASTRAL and CONCAT trees both reveal a relatively short branch connecting the most recent common ancestors of Archaea and Bacteria (Figs. 1, 4a, c; Supplementary Fig. 17). Its length is fractional comparing with the dimensions of both clades (appr. 0.13–0.14 by conserved sites, 0.09–0.11 by random sites) (Fig. 4c, e; Supplementary Table 4). This pattern is in contrast to previous trees built using fewer marker genes, all or most of which are ribosomal proteins formerly considered to be effective markers for assessing global microbial evolution[22] (e.g.[13,19,30]). To further test how the choice of marker genes affects the inter-domain distance, we estimated branch lengths of the ASTRAL tree using 30 ribosomal proteins extracted from the genomes. Consistent with previous studies, we observed an elongated branch connecting Bacteria and Archaea. Its length relative to clade dimensions (1.0–1.6) is about ten times the

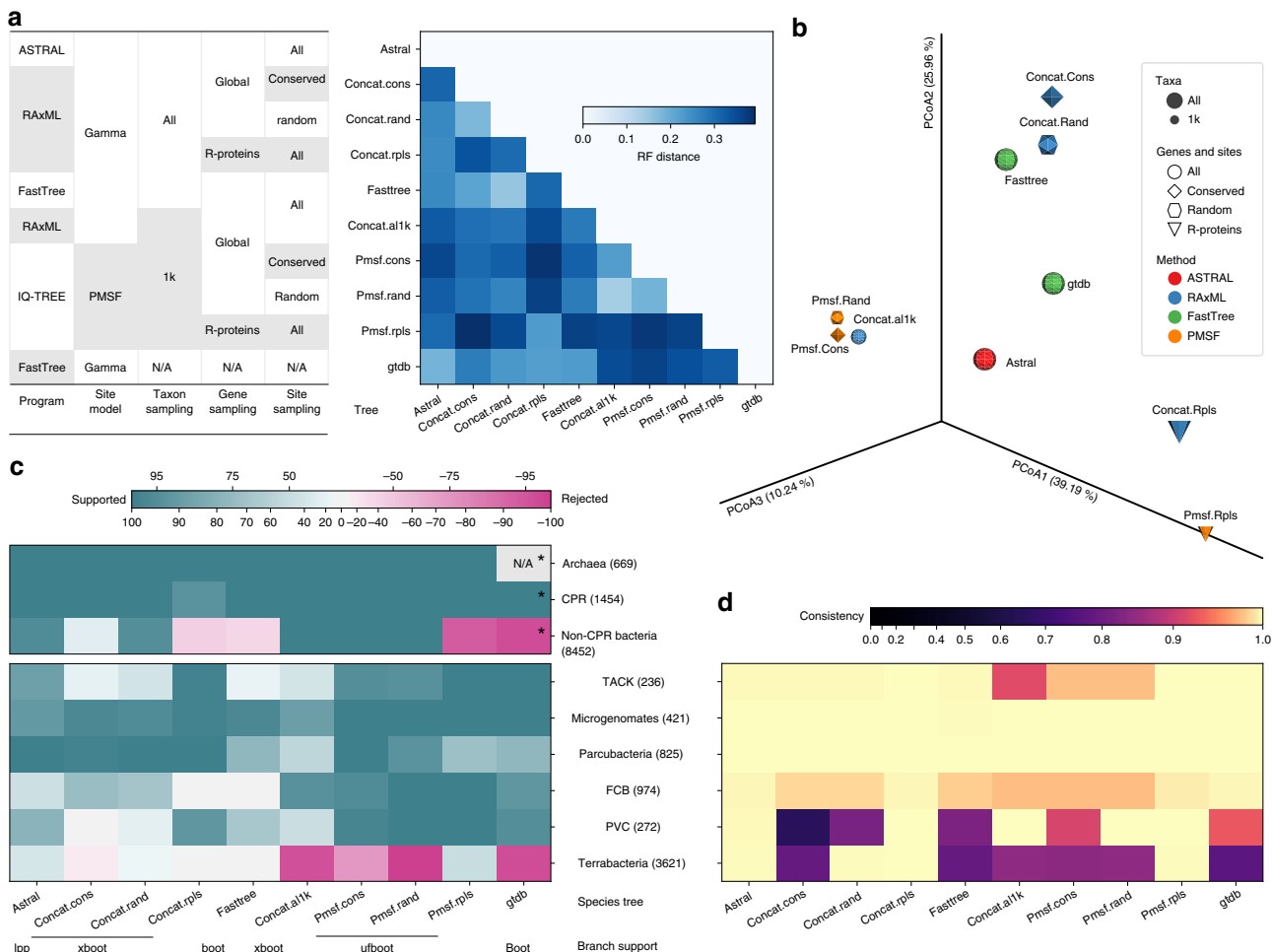

**Fig. 3** Comparison of topologies of multiple species trees. Nine species trees reconstructed in this study, plus a previously published tree, GTDB release 86.1 were cross-compared. The methods for building those trees are summarized in the inset table. **a** Matrix of normalized Robinson–Foulds (RF) distance, which measures the overall topological discrepancy between two trees based on the shared taxa. **b** PCoA of the RF distance matrix. **c** Branch support-informed support/rejection degrees (see the Methods section) for the monophyly of Archaea, CPR, and non-CPR Bacteria, and six super phyla defined by NCBI. Clades were defined according to the tax2tree auto-curation result based on each tree. Note that super phyla Asgard (eight taxa) and DPANN (five taxa) are not shown due to low taxon representativeness, but are discussed in Supplementary Note 5. Numbers in parentheses indicate the number of taxa under each clade according to the ASTRAL tree. The types of branch support values are indicated below tree names: "lpp": local posterior probability, "boot": classical bootstrap, "xboot": rapid bootstrap, "ufboot": ultrafast bootstrap. Note that different branch support types cannot be directly compared. Note (*) that the GTDB phylogeny consists of two trees independently built for Archaea and Bacteria, respectively, thus the support/rejection for Archaea cannot be assessed. Meanwhile, the bacterial tree was rooted using a midpoint strategy, making Chloroflexi the outgroup to all remaining bacteria (including CPR). But it should not be considered as a rejection to the monophyly of non-CPR Bacteria in the sense of evolutionary relationships. **d** Consistency between NCBI-defined super phyla and tree topologies, computed using tax2tree (see Methods). Source data are provided as a Source Data file.

estimate using the 381 global marker genes (Fig. 4b, d, e; Supplementary Table 4). We also calculated the overall phylogenetic distance between taxa of the two domains, as relative to the intra-domain distances. This relative distance based on the ribosomal proteins (4.5–5.0) is around three times that of the distance by the global marker genes (1.5–1.6) (Fig. 4f; Supplementary Table 4).

Considering the special status of CPR, we performed an independent test with the 1454 CPR genomes removed from the data set prior to de novo phylogenetic inference, and we compared the results to the main results (Fig. 4e, f) with the CPR clade pruned from the tree. These trees continued to reveal the substantially shorter branch and tip-to-tip distances between the two domains as recovered by using the 381 global marker genes as compared with using the 30 ribosomal proteins (Supplementary Fig. 18, Supplementary Table 5).

We tested whether the potential saturation of amino acid substitution could cause an underestimation of the domain separation. The ratio between phylogenetic distance and sequence distance is similar between pairs of taxa selected both from Bacteria, both from Archaea, or one from each domain (Supplementary Fig. 19). This indicates that the relative length of the branch connecting the two domains compared with the intra-domain branches is not substantially impacted by saturation.

We further evaluated how individual gene trees impact the observed proximity between Bacteria and Archaea. Except for a few outliers, which include several "core" genes like *rpoC* (RNA polymerase subunit β', 18.27), *tuf* (elongation factor Tu, 12.18), and *fusA1* (elongation factor G, 9.54), most gene trees have relative Archaea–Bacteria distances between 1 and 3 (mean: 2.00) (Fig. 5a, b), which is consistent with that of the species tree

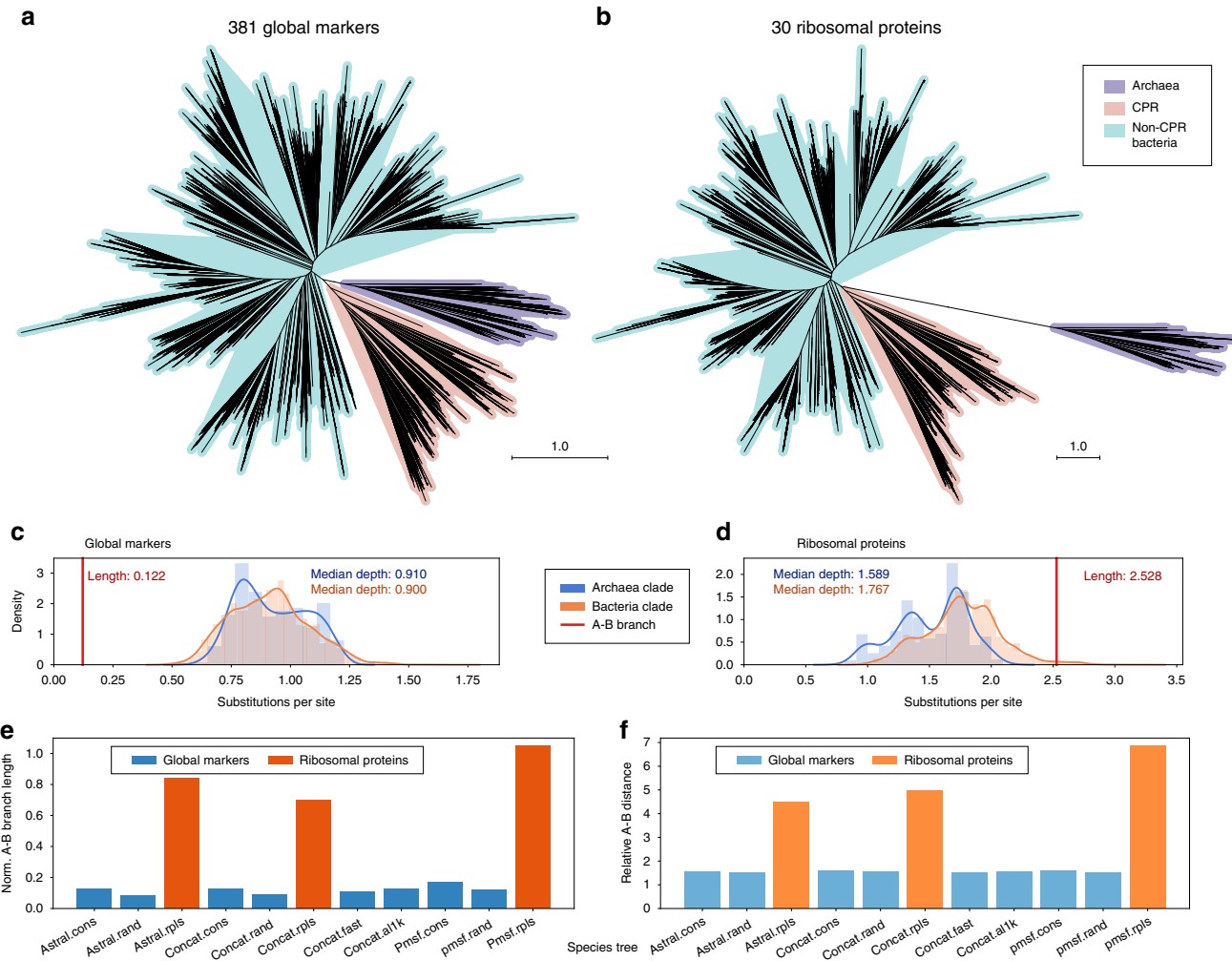

**Fig. 4** Evolutionary proximity between domains Archaea and Bacteria. **a**, **b** The unrooted, drawn-to-scale ASTRAL tree with branch lengths estimated using the 381 global marker genes (conserved site sampling) (**a**) or using the 30 ribosomal proteins (**b**) are displayed, with color codes highlighting domain-level relationships (Archaea and Bacteria, the latter of which consisting of CPR and non-CPR Bacteria). Scales are in the unit of the number of substitutions per site. **c**, **d** For each domain (blue: Archaea, orange: Bacteria), the histogram and Gaussian kernel density function of the depths of all descendants (sums of branch lengths from a tip to the lowest common ancestor (LCA) of the clade) are plotted, with the median depth displayed; the length of branch connecting the LCA of A(rchaea) and the LCA of B(acteria) is marked as a red vertical line, with its value displayed. **e**, **f** Evolutionary distance between domains Archaea and Bacteria by multiple trees (tree names follow Fig. 3), among which the branch lengths of the ASTRAL tree were estimated using differential gene and site samplings, respectively. Two metrics are displayed: the A–B branch length normalized by the median depth of all tips in the tree (**e**), and the relative distance between Archaea taxa (tips) and Bacteria taxa (**f**, see the Methods section). Color codes highlight evolutionary distances indicated by differential gene samplings (blue: the 381 global marker genes, orange: the 30 ribosomal proteins). Source data are provided as a Source Data file.

summarizing the global marker genes, and in contrast to that obtained using only the ribosomal proteins (Fig. 5a).

**Heterogeneity among individual genes' evolutionary histories.** Because microbial genomes are highly dynamic and prone to HGTs, it is important to investigate the discrepancies among the evolutionary paths of individual gene families to better understand the evolution of genomes[20]. To measure the topological concordance between two trees, we used the quartet score[31], which correlates well with the traditional Robinson–Foulds (RF) metric (Supplementary Fig. 20c)[32], resulting in a distribution of gene trees tightly centered around the species tree (Fig. 5d; Supplementary Figs. 20a, 21).

The discordance between the 381 single-gene trees and the species tree varied widely (Fig. 5c). The quartet scores (larger is more similar, with identical trees scoring 1.0) ranged from 0.372 (*cmpD*) to 0.943 (*hslU*), with the mean and standard deviation

being 0.653 ± 0.136. Many of the individual trees with high similarity to the species tree belong to genes involved in the core machinery of genetic information processing, such as those encoding DNA/RNA polymerase subunits, ribosomal proteins, and elongation factors, while genes involved in peripheral functions such as membrane transport are frequently more distant from the species tree (Fig. 5c, d). This pattern is generally consistent with a previous study on a small taxon set[20]. While determining the cause of discordance for individual genes is beyond the scope of this study, the pattern we observed is consistent with a reduced rate of HGT for fundamental genes compared with those with less conserved functional significance[33]. There was no apparent correlation between a gene tree's concordance with the species tree and the prevalence of the gene in the sampled genomes (Supplementary Fig. 20d, e), suggesting that universality is not necessarily indicative of fidelity.

To further test the impact of gene tree discordance on the species tree, we sequentially removed genes from the low end of

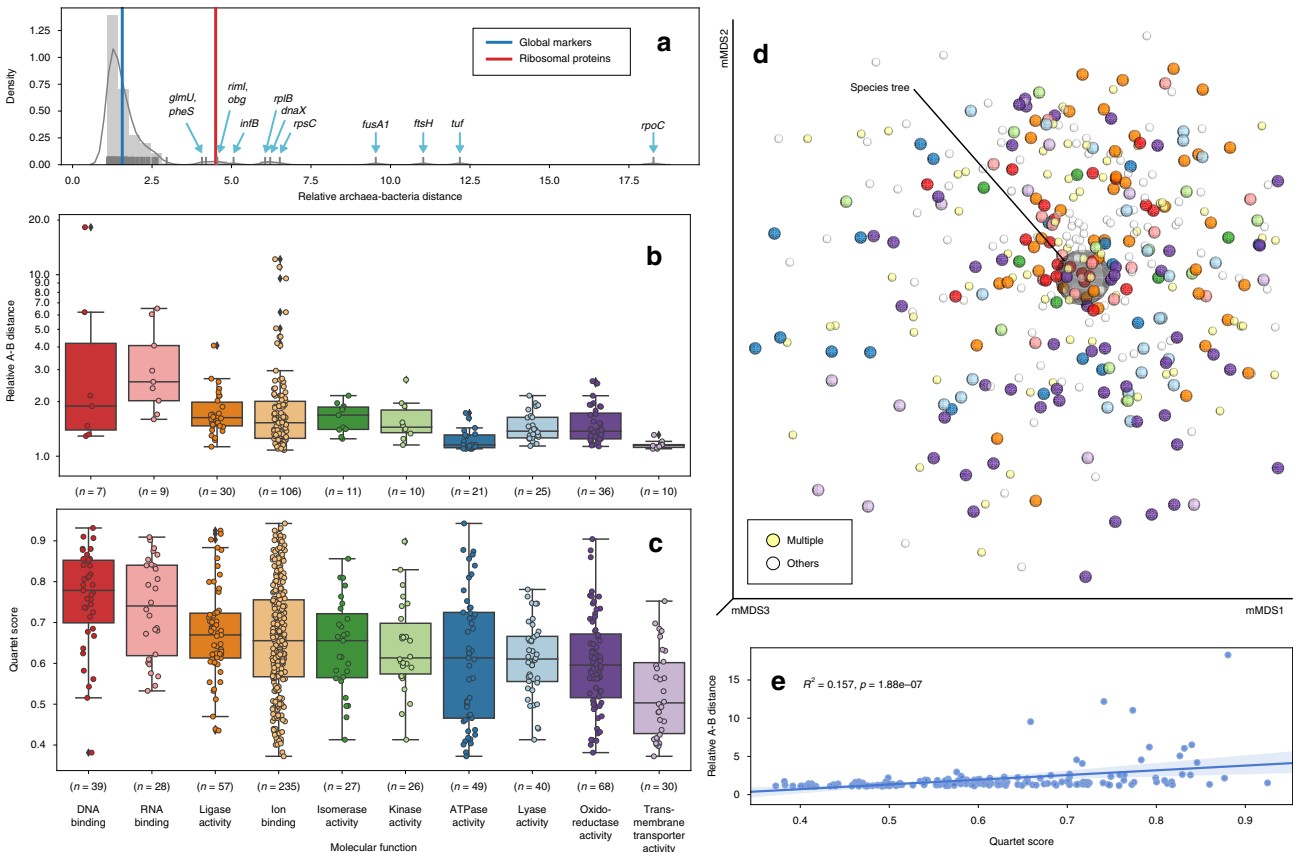

**Fig. 5** Relative Archaea–Bacteria distances indicated by individual gene trees and their concordance with the species tree. **a** Distribution of relative A (rchaea)–B(acteria) distances of individual gene trees. A total of 161 gene trees are shown, selected such that both domains have at least 50% taxa represented in each tree. A histogram with Gaussian kernel density function and a rug plot representing individual data points are displayed. The blue and red vertical lines indicate the values of the ASTRAL species tree with branch lengths estimated using the global markers and the ribosomal proteins, respectively. Gene names are labeled at data points separated from the main cluster. **b** Distribution of relative A–B distances by functional category (GO slim term under the "molecular function" master category) of the 161 gene trees. The top ten most frequently assigned categories in all gene trees are shown. Boxplot components: center line, median; box limits, upper and lower quartiles; whiskers, 1.5 × interquartile range; black diamonds, outliers. **c** Distribution of quartet scores of all 381 gene trees vs. the species tree. Boxplot components are identical to **b**. **d** mMDS plot of the quartet distance matrix of all 381 gene trees plus the species tree (semi-transparent big gray ball in the center). The color scheme for genes annotated by exactly one of the top ten functional categories (normal-sized balls) is consistent with **b** and **c**, except that the category "ion binding" is omitted due to its high frequency. Genes annotated by more than one of the top ten categories (light yellow), or by categories other than the top ten (semi-transparent light gray) are indicated by smaller balls. **e** Linear regression of relative A–B distances vs. quartet scores of the 161 gene trees. The squared Pearson correlation coefficient ($R^2$) and two-tailed p-value are displayed. Source data are provided as a Source Data file.

the quartet score rank (Supplementary Note 2). ASTRAL produced stable topologies in this test (Supplementary Fig. 22a–c). We next tested the impact on phylogenetic distances. There was a weak positive correlation (Pearson correlation $R^2 = 0.157$, $p = 1.88e{-}07$) between the quartet score and the relative Archaea–Bacteria distance (Fig. 5e). When the branch lengths of the species tree were estimated using genes with high quartet scores only, the distance moderately increased, yet remained far from the result by using the ribosomal proteins (Supplementary Table 6). This suggests that non-vertical transmission of genetic information has only a limited impact on our updated estimates of the inter-domain distance.

**Heterogeneity across sites.** Inferring phylogenetic trees at deep time scales, beyond the heterogeneity of gene histories, requires paying attention to the heterogeneity of substitution processes across the genome[34,35]. As recently as 2015, Gouy et al. declared the jury to still be out on the root of the tree of life[36], partially due to difficulties in modeling heterogeneity of sequence evolution across sites. In particular, changes in amino acid frequency across

sites of the same gene can exacerbate long-branch attraction[37]. To account for these difficulties, we tested whether our main conclusions stand if the data are analyzed with a recently developed model, PMSF, which considers heterogeneity in the amino acid substitution process[38] (Fig. 3: "pmsf.cons" and "pmsf.rand"). Because of the computational complexity of this approach, we had to limit these analyses to 1000 taxa. At this sampling depth, we were also able to build a tree using all sites and the CONCAT method for comparison ("concat.al1k").

The topology of PMSF trees largely resembled the RAxML trees with the same taxon sampling (Fig. 3a, b). The impact of using the PMSF model instead of site homogeneous models on the topology and branch lengths was small compared with the impact of taxon, locus, and site sampling (Supplementary Fig. 23, Supplementary Note 2). The PMSF trees continued to support a large portion of relationships among deep branches recovered by the full-scale trees (Fig. 3c, d; Supplementary Fig. 3). The evolutionary proximity between Bacteria and Archaea continued to hold with the PMSF trees. Meanwhile, the PMSF tree based on ribosomal proteins ("pmsf.rpls") also resembled the corresponding full-scale tree in

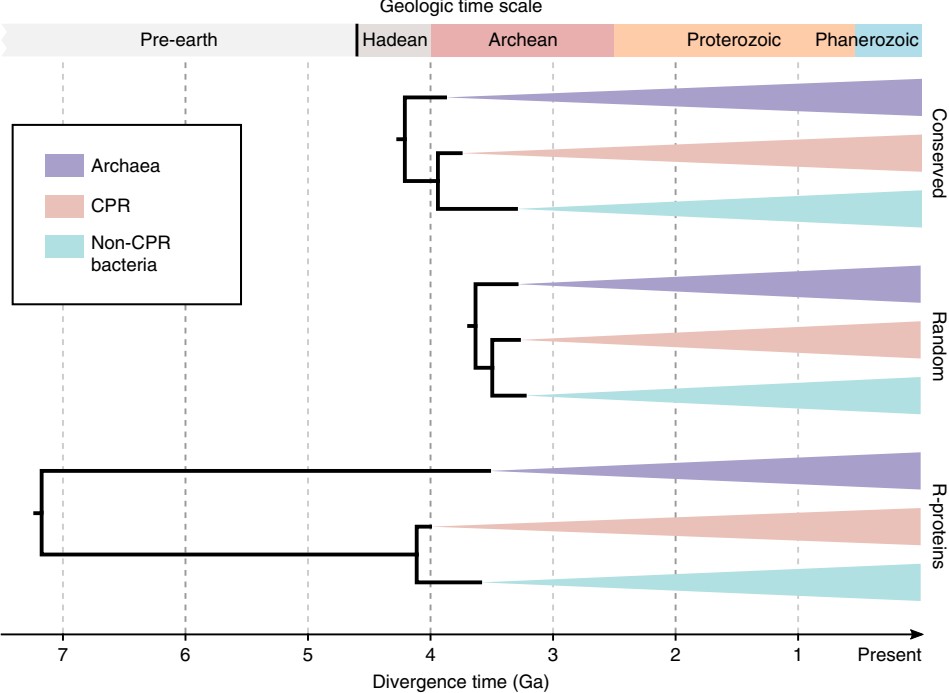

**Fig. 6** Estimated ages of basal diversifications. Divergence time estimation was performed using maximum likelihood based on the ASTRAL tree topology with branch lengths estimated using most conserved or randomly selected sites from the 381 global marker genes, or using the 30 ribosomal proteins. A universal clock was assumed. The Cyanobacteria/Melainabacteria split was constrained to 2.5–2.6 Ga. The best estimates from ten technical replicates per setting are displayed (see Supplementary Table 8). Details and alternative results of divergence time estimation are elaborated in Supplementary Note 6. Source data are provided as a Source Data file.

suggesting a long distance between Bacteria and Archaea (Fig. 4e, f; Supplementary Fig. 24, Supplementary Table 7). Taken together, this shows that our phylogenies and main conclusions are robust when considering site heterogeneity.

## Discussion
The origin and evolution of life have been among the most intriguing scientific questions, with the current widely adopted notion being the three-domain system: Bacteria, Archaea, and Eukaryota[2]. Recent phylogenomics studies typically indicated a long distance between Bacteria and Archaea, with Eukaryota as an ingroup of the Archaea clade[10,11]. In this work, we built a reference phylogeny of over 10,000 bacterial and archaeal genomes, covering a significant proportion of the known biodiversity with available genome data. The result provides an updated view of microbial evolution, showing that Bacteria and Archaea, the two microbial domains conventionally but controversially grouped by the term "prokaryotes"[39], are much closer in evolutionary proximity than estimates using a smaller number of "core" genes, such as the ribosomal proteins. This observation was further supported by extensive analyses using multiple tree-building methods, with consideration of taxon and site sampling, amino acid substitution heterogeneity and saturation, and non-vertical evolution, and was robust against the exclusion of CPR taxa. Interestingly, applying a simple universal molecular clock as well as relaxed clock rates to date, our trees resulted in divergence time estimates of major lineages that are compatible with geological timeline only when using the global markers, but not when trees are restricted to ribosomal proteins (Fig. 6; Supplementary Figs. 25, 26, Supplementary Tables 8-10, see full details in Supplementary Note 6). These comparisons suggest accelerated evolution in ribosomal proteins during the separation between Bacteria and Archaea. They show the limitation of using core

genes alone to model the evolution of the entire genome, and highlight the value in using a more diverse marker gene set.

Our work highlights the value of even taxon sampling, a global marker gene set representing the larger average of genome content, and comparative phylogenetic analyses. These procedures largely reduced the bias of gene choice in exploring genome evolution, and allowed us to characterize the evolutionary discrepancies of individual gene families. Despite these efforts, some lineages are still underrepresented in our sampling, such as DPANN[4], which has genomes that are often missing many of the 381 marker genes (detailed in Supplementary Note 5). Moreover, the rapid growth of genomic data has led to the absence of some newly discovered groups from our tree. While it is impractical to repeat all of our analyses to include all new genomes, it is of interest to assess whether the newly discovered microbial diversity may impact our results. Prior to submission of this article, we updated the genome collection from NCBI on May 23, 2019, and selected 187 new genomes representing phyla as defined by the newest NCBI and GTDB taxonomies that are absent or underrepresented in the current set of 10,575 genomes (see the Methods section). Phylogenetic trees built using the extended genome set continued to support the domain-level relationships in both topology and evolutionary distance as recovered by the main analysis (Supplementary Fig. 27, Supplementary Table 11, see Supplementary Note 7 for full details). Finally, we note that the inclusion of eukaryotes is challenging with the current marker gene set due to the overall sparsity of detectable homology. Further improvements in methodology are important in order to deliver a robust phylogeny that encompasses all forms of life.

## Methods
**High-performance computing**. Analyses of the genome data sets in this study were computationally intensive. Heavy computations used the "Comet" supercomputer located at the San Diego Supercomputer Center (SDSC). Each standard

node is equipped with 24 Intel Haswell CPU cores and 128 GB of DDR3 memory, while each GPU node is equipped with four NVIDIA P100 GPUs, plus 28 Broadwell CPU cores and 128 GB memory. A small proportion of the computations used the "Barnacle" computer system operated by the Knight Lab, each node of which has 32 Haswell CPU cores and 256 GB DDR3 memory. Whenever possible, all CPU cores were used in a typical multi-processing task to minimize run time. Tasks that required >128 or 256 GB memory used the large-memory nodes of Comet, featuring 64 CPU cores and 1.5 TB memory per node. Benchmarking of the prototype selection algorithms and some local developments were performed on the "WarpDrv" workstation, equipped with 32 Intel Sandy Bridge CPU cores and 256 GB of DDR3 memory.

**Retrieval of genome data and metadata**. Microbial genomes were downloaded from the NCBI genome database (GenBank and RefSeq) as of March 7, 2017. We used and provided updates related to this work to the automated workflow RepoPhlAn (https://bitbucket.org/nsegata/repophlan, commit 03f614c) to download genomes from the NCBI server. Each genome was given a unique identifier, which was derived from the NCBI accession of the corresponding assembly, but without version number. For example, a genome with assembly accession "GCF_000123456.1" was identified as "G000123456" in this study. In cases when the same genome was present in both GenBank (accession starting with "GCA_") and RefSeq (accession starting with "GCF_"), only the RefSeq version was kept.

**Annotation and classification of marker genes**. The functional annotation of the 400 PhyloPhlAn marker genes[25] was performed by aligning the protein sequences of the 400 marker genes (inferred from 2887 bacterial and archeal genomes as described in Segata et al.[25]) against the UniRef50 database (March 2018 release) using BLASTp. The best hit for each gene was taken and queried against the UniProt database for gene and protein names. To categorize genes by function, the UniProt entries were translated into Gene Ontology (GO) terms[40] with the "subset_prokaryote" tag (March 2018 release). Because not all UniProt entries have corresponding GO terms, manual curation was involved to pick the most appropriate GO terms for those cases by examining the BLAST hit table. GO terms were further translated into GO slim terms to obtain higher-level functional categories. Note that this analysis is independent from the phylogenetic analysis of the current genome data set, and the result can be used as a reference for PhyloPhlAn users.

**Analyses of genome sequences and identification of marker genes**. The DNA sequences of the 86,200 bacterial and archaeal genomes were subjected to the following analyses:

1. The quality scores for DNA, protein, tRNA, and rRNA were calculated following Land et al.[41].
2. A MinHash sketch was built for each genome using Mash 1.1.1[42], with default settings (sketch size = 1000, k-mer size = 21), based on which a pairwise distance matrix was built for the entire genome pool. In brief, MinHash is a k-mer hashing technique that enables the quantification of genome-to-genome distance. It is efficient for very large genome sets, and it correlates well with the conventionally used average nucleotide identity (ANI)[42].
3. Although NCBI provides genome annotations, we chose to re-annotate the genomes using a uniform protocol to ensure consistency. Specifically, open-reading frames (ORFs) were predicted using Prodigal 2.6.3[43], in the single-genome mode, and allowing ORFs to runoff edges of scaffolds.
4. Based on the predicted ORFs, the 400 marker genes were inferred and extracted using the phylogenomics pipeline PhyloPhlAn (commit 2c0e61a)[25], in which the 400 marker genes were originally introduced. In brief, we used USEARCH v9.1.13 to align ORFs against the reference marker gene sequences (see above) at an E-value threshold of 1e-40. It then selected the highest bit score hit of each ORF. Should more than one hit per marker per genome was observed, the highest bit score hit was selected as the representative of that marker gene.
5. The completeness, contamination, and strain heterogeneity scores were computed using CheckM 1.0.7[44] with the default protocol ("lineage_wf").

**Prototype selection and genome sampling**. Proper taxon sampling is a key prerequisite to inferring an unbiased view of organism evolution[45,46]. Beyond computational challenges in robust tree-building, the highly uneven distribution of known biodiversity (e.g., 40.0% of all genomes (34,507) belong to the nine most-sequenced species) requires deliberate subsampling to reduce the bias from the resulting phylogeny in representing a global view of evolution. We therefore applied the data-reduction strategy of "prototype selection"[47], which subsamples genomes from the pool such that they represent the largest possible biodiversity—in terms of maximized sum of pairwise distances as defined by k-mer signatures (Supplementary Fig. 1a). We developed a heuristic (detailed in Supplementary Note 1), capable of handling the size of the current genome pool, with results comparable with or better than published alternatives (Supplementary Fig. 1b–e).

Using this algorithm and by applying multiple criteria, we downsampled the 86,200 bacterial and archaeal genomes to 11,079. The procedures are detailed below.

1. Excluded genomes with marker gene count < 100 or contamination > 10%. The marker gene count threshold 100 was chosen because it is sufficiently large to yield high resolution of the tree using ASTRAL (Supplementary Fig. 10a, c). The contamination threshold 10% is inline with the medium- and low-quality draft genome standards proposed by Bowers et al.[6]. Nevertheless, we did not adopt the completeness and tRNA/rRNA coverage thresholds[6], because the 400 protein-coding marker genes are more relevant for phylogenetic reconstruction.
2. Included the NCBI-defined reference and representative genomes (https://www.ncbi.nlm.nih.gov/refseq/about/prokaryotes/).
3. Included genomes that are the only representative of each taxonomic group from phylum to genus.
4. Included genomes that are the only representative of each species without defined lineage (no classification other than species).
5. Executed the prototype selection algorithm developed in this work: "destructive_maxdist" (see Supplementary Note 1) based on the distance matrix defined by MinHash signatures, with the already included genomes as seeds, to obtain a total of 11,000 genomes.
6. For each phylum to genus, and species without classification from phylum to genus, selected one with the highest marker gene count. This added 79 genomes to the selection.

These 11,079 genomes were subjected to our phylogenetics protocol, during which further filtering was performed based on sequence alignment quality (see below). Eventually, 10,575 genomes were retained.

**Impact of alternative genetic codes**. We chose to uniformly apply the standard archaeal and bacterial genetic code table 11 to all genomes in order to minimize bias. Reports have shown that several lineages, such as Mycoplasma/Spiroplasma[48], Hodgkinia[49], and Absconditabacteria[50], use alternative genetic code tables 4, 25 and others, in most of which a stop codon is repurposed to encode for an amino acid, resulting in ORF elongation. We did not incorporate alternative genetic codes, however, because there is no accurate way to associate each of the 86,200 genomes with its true genetic code. Incorrect truncation of ORFs may unnecessarily exclude genes and taxa, whereas incorrect elongation of ORFs could result in artificially long branches, because the amino acid sequence after a true stop codon is likely relaxed from selective pressure. Considering our goal of inferring phylogenetic topology and distances, we decided to only use the standard genetic code.

However, we did test the impact of using alternative genetic code on the gene and taxon sampling. We ran Prodigal 3.0.0-rc1, which automatically switches from genetic code 11 to 4 if the average ORF length is too short. This resulted in altered gene calling results in 453 out of the 86,200 genomes, of which 63 had overly short ORF lengths even when using genetic code 4. PhyloPhlAn marker gene discovery on the other 390 genomes with genetic code 4 suggested marginal increase in the extracted number of the 400 marker genes per genome (1.23 ± 5.28, mean and std. dev.). Only seven additional genomes which had <100 marker genes managed to pass this threshold (see above) after switching to genetic code 4. Therefore, omitting alternative genetic code has little impact on the inclusion of genomes.

**Metric multidimensional scaling (mMDS) of genome distances**. The effect of prototype selection was visualized using the mMDS technique, which renders a low-dimensional plot that minimizes the loss of information when transforming from the high-dimensional data. We performed mMDS using the "mds" function implemented in scikit-learn 0.19.2[51] on the genome distance matrix, using the default setting, to compute the coordinates at the top five axes. The resulting coordinates were visualized with the interactive tool Emperor[52] as bundled in QIIME 2 release 2017.12[53].

**Protein sequence alignment and filtering**. Protein sequences of each of the 400 marker gene families were aligned using UPP v2.0[54], a phylogeny-based and fragmentary-aware alignment tool. UPP consists of several sequentially connected modules. It first identifies suspected fragmentary sequences, then calls PASTA v1.8.0[55] to align the remaining sequences and build a phylogeny (backbone tree) based on them. Then it builds an ensemble of HMMs using HMMER[56] based on the phylogeny. Finally, it aligns the fragmentary sequences to the HMMs and selects the one with the best match. Sequences that are 25% longer or shorter than the median sequences were considered as fragments and excluded from the backbone. More specifically, PASTA first builds a starting tree, performs a tree-based clustering of the sequences, and builds a spanning tree from these clusters. Then it calls MAFFT v7.149b[57] to align the sequences in each cluster, and calls OPAL[58] to merge the alignments of adjacent clusters according to the spanning tree, and finally uses transitivity to perform the subsequent merging. To ensure the quality of the alignment, we filtered out extremely gappy sites and sequences: sites with >90% gaps were deleted from the alignments, followed by the dropping of sequences with >66% gaps.

**Filtering of marker genes**. To ensure the quality of the species tree built upon these marker genes[59], we filtered out the genes that were not aligned reliably by UPP. As such, the marker genes with >75% gaps in the aforementioned alignments were excluded from the pool, leaving 381 marker genes in total. The threshold 75% was chosen based on the distribution pattern of per-gene alignment quality (Supplementary Fig. 2d).

**Filtering of outlier taxa from gene trees**. We removed suspected outliers by detecting the taxa on disproportional long branches and filtering them out from the phylogeny inferred by FastTree[27]. To do this, we applied TreeShrink[60] v1.1.0, a method that simultaneously detects long branches on a set of gene trees by identifying a set of taxa that could be removed from each gene so that the gene trees are maximally reduced in diameter. We used FastTree 2.1.9 to infer preliminary gene trees of the 381 selected genes, then ran TreeShrink to detect outlier long branches in these trees, with the per-species test with α = 0.05 (5% false-positive tolerance). Finally, we dropped genomes that contained <100 marker genes post gene tree filtering.

**Gene tree reconstruction**. Gene tree topologies were reconstructed using the maximum likelihood (ML) method as implemented in the state-of-the-art phylogenetic inference program RAxML 8.2.10[26]. The best amino acid substitution model for each of the 381 universal marker genes was inferred using RAxML's built-in script ProteinModelSelection.pl. Three phylogenetic trees were reconstructed for each gene family: one using a starting tree computed by the fast ML approach implemented in FastTree) and the other two using parsimony trees built with random seeds 12345 and 23456. RAxML was executed with the ML search convergence criterion (-D) and the CAT rate heterogeneity model without final optimization (-F) to reduce the execution time.

For each of the 1143 topologies (3 × 381), another RAxML run was executed to optimize branch lengths and to compute likelihood scores under the robust, but expensive Gamma rate heterogeneity model. Because of numerical instability, at least one of the RAxML runs failed for 39 of the 381 gene families. For those cases, IQ-TREE 1.6.1[61], an alternative and faster maximum likelihood program, was used instead to optimize branch lengths using the same model (G4). The tree with the highest likelihood score among the three runs was retained for downstream applications. In 161 gene families, this tree was from the run with the FastTree starting tree, while in the remaining gene families the best tree was from either one of the random seeds.

**Species tree reconstruction by summarizing gene trees (ASTRAL)**. A species tree was reconstructed by summarizing the 381 gene trees, using ASTRAL-MP[62] (implementing ASTRAL-III algorithm[28]) 5.12.6a. This analysis was run on the Comet supercomputing cluster using 24 cores and 4 GPU acceleration. In the resulting tree, the branch lengths represent the units of coalescence. Each branch has three support values: (1) effective number of genes (EN): the number of gene trees that contain some quartets around that branch; (2) quartet score (QT): proportion of the quartets in the gene trees that support this branch; (3) local posterior probability (LPP): the probability this branch is the true branch given the set of gene trees (computed based on the quartet score and assuming incomplete lineage sorting (ILS))[31].

**Branch length estimation for the ASTRAL tree**. The branch lengths of a summary tree generated by ASTRAL are in coalescent units and only for internal branches. In order to obtain "conventional" branch lengths, i.e., the expected number of amino acid substitutions per site, we ran IQ-TREE using the concatenated alignment (most conserved or randomly selected sites as described below) as input, the ASTRAL tree as the topological constraint, and the LG + Gamma model. Branch lengths obtained using both site categories were highly correlated (Supplementary Note 2).

**Species tree reconstruction based on the concatenated alignment (CONCAT)**. The alignments of the 381 marker genes were concatenated into a supermatrix. Due to the computational challenge in running classical maximum likelihood tree reconstruction on the full-scale data set, we had to downsample it to around 38 k amino acid sites. In order to explore the impact of site sampling on tree topology and branch lengths, we separately adopted two strategies for site sampling: (1) selected up to 100 most conserved sites per gene. The degree of conservation was estimated using the "trident" metric[63], which is a weighted composition of three functions: symbol diversity, stereochemical diversity, and gap distribution. The PFASUM60 substitution matrix was used for computing the stereochemical diversity[64]. (2) randomly selected 100 sites per gene from sites with <50% gaps.

For the downsampled supermatrix, a maximum likelihood tree was first built using FastTree, with LG model for amino acid substitution and Gamma model for rate heterogeneity. Using this FastTree tree as the starting tree, plus two maximum parsimony trees generated from random seeds (12345 and 23456), we performed three independent runs using RAxML, with the LG + CAT models (PROTCATLG), with rapid hillclimbing (-f D) and without final Gamma optimization (-F). With the resulting topologies, we performed branch length optimization and likelihood score calculation using IQ-TREE, with the LG +

Gamma models (LG + G4). We further performed 100 rapid bootstraps using RAxML to provide branch support values.

**Species tree reconstruction based on ribosomal proteins**. To test the impact of choice of marker gene set on the topology and relative distances among major taxonomic groups, we conducted a separate analysis in which the species tree was built using ribosomal protein sequences. We identified and extracted 30 ribosomal protein families using the program PhyloSift 1.0.1[65] with its marker database released on August 8, 2017. If more than one copy of a marker protein was detected in a genome, all copies were discarded. After this filtering, genomes with fewer than 25 marker proteins were dropped from the data set, resulting in a total of 9814 genomes of the original 10,575. Sequences of each ribosomal protein family were aligned using UPP as described above. The resulting alignments were concatenated and subjected to RAxML tree reconstruction using the LG model for amino acid substitution[66] (which is the best model for 304 out of the 381 genes based on RAxML's model selection) and the CAT model for rate heterogeneity (PROTCATLG). The resulting tree was then fed into IQ-TREE for branch length optimization, with the Gamma model for rate heterogeneity. One hundred rapid bootstraps were executed in RAxML to provide branch support.

The same concatenated alignment was also used to estimate the branch lengths for the ASTRAL tree based on the 381 marker gene trees. Because the quality of an ASTRAL tree improves as the number of gene trees increases (Supplementary Fig. 10a, c), running ASTRAL on only 30 trees of structurally and functionally highly related genes is of limited value. Thus we decided to not to run ASTRAL de novo, but only to assess the impact of ribosomal proteins on the branch lengths of the existing ASTRAL tree.

**Species tree reconstruction and branch length estimation with CPR taxa excluded**. We followed the same protocol as stated above to reconstructed species trees and estimate branch lengths based on the protein sequence alignments with the 1454 CPR taxa removed, leaving 9121 taxa. Only one modification was made to the main protocol in order to reduce the computational expense for reconstructing the 381 gene trees: Instead of running RAxML three times per gene and selecting one tree with the highest Gamma likelihood, we ran RAxML once per gene using the random seed 12345. The two alternative site sampling schemes: most conserved ("cons") and randomly selected ("rand") as demonstrated in the main result were both tested, using the same amino acid sites as in the main protocol in each scheme.

**Species tree reconstruction using site heterogeneous models (PMSF)**. We built alternative CONCAT trees using the posterior mean site frequency (PMSF) method[38] implemented in IQ-TREE, which considers mixture classes of rates and substitution models (here the LG model) across sites. Because this method is computationally expensive, we downsampled the 10,575 taxa to 1000 (see below for the taxon downsampling strategy). ModelFinder (which is part of IQ-TREE)[67] was used to select an optimal model among the empirical profile mixture models C10 to C60[68], plus the site homogenous model (with Gamma rate across sites) as a control. This analysis consistently chose C60 as the optimal model for all tests. Therefore, we used the LG + C60 model for PMSF phylogenetic reconstruction. PMSF requires a guide tree, which we obtained from ModelFinder results. Computational challenge limited this analysis to at most 1000 taxa (which consumed 1.43 TB memory, close to the 1.5 TB physical memory equipped in our high-memory nodes). Branch support values were computed using the ultrafast bootstrap (UFBoot)[69] method implemented in IQ-TREE. In parallel to this analysis, we performed phylogenetic inference using the Gamma model ( + G) or the Free-Rate[70] model (+R) on the same 1000-taxon input data for comparison purpose.

**Species tree reconstruction using implicit methods**. We applied two implicit strategies for inferring the evolutionary relationships among the sampled genomes. They are not based on the alignment of homologous features across multiple genomes, but instead are based on the predefined distances among genomes. Specifically, they are the Jaccard distances defined by the MinHash signature (see above), and by the presence/absence of the 400 marker genes (see above). The conventional neighbor joining (NJ) method as implemented in ClearCut 1.0.9[71] was used to reconstruct phylogenetic trees from the two distance matrices, respectively.

**Rooting and post-manipulation of species trees**. We rooted the species tree at the branch connecting the Archaea clade and Bacteria clade, according to the widely adopted hypothesis of life evolution[72–74]. The absence of Eukaryota does not impact the placement of root, since Eukaryota is considered derived, as a sister group or ingroup of Archaea in this hypothesis. We want to remind readers that this hypothesis is not without controversy[75,76]. The discovery and study of CPR and other divergent or transitional groups may provide materials for a second examination of this hypothesis, although this is beyond the scope of this study.

Internal nodes were flipped to follow the descending order (i.e., child nodes are sorted from less descendants to more descendants). Incremental numbers were assigned to internal node IDs in a pre-order traversal of the tree starting from the root (i.e., root = N1, LCA of Archaea = N2, LCA of Bacteria = N3, etc.). These

node IDs can be used as unique identifiers in downstream analyses and applications.

**Phylogeny-based downsampling of taxa**. We designed a protocol to downsample taxa from the 10,575 genomes for further phylogenetic analyses. We adopted the relative evolutionary divergence (RED) metric[14], as the core of our subsampling strategy. This metric allowed us to select large clades that best represent the deep phylogeny. Specifically, we calculated RED for all nodes (terminal and internal) of the ASTRAL tree (i.e., the tree shown in Fig. 1) using TreeNode functions implemented in scikit-bio 0.5.2[77]. Nodes were selected iteratively from the low end of the RED list, with ancestral nodes (if any) of the current node dropped from the selection at each iteration, until the desired number of clades $n$ was achieved.

Within each selected clade, four criteria were sequentially applied to the descendants until one taxon was selected: (1) contains the most marker genes; (2) contamination level is the lowest; (3) DNA quality score is the highest; (4) random selection (if there were still more than one taxon after applying the other three criteria). This protocol guaranteed the selection of $n$ taxa, which maximize the representation of deep phylogeny.

**Visualization and annotation of trees**. Unique colors were assigned to selected NCBI-defined taxonomic groups above phylum, and phyla with 100 or more representatives in the sampled genomes. Colors of taxa were directly assigned based on their NCBI taxonomy assignment. Colors of clades and branches were determined based on the tax2tree decoration. The trees were rendered using iTOL v4[78] (unrooted or circular layouts) or FigTree 1.4.3[79] (rectangular layout).

**Comparison of multiple trees**. We used both the classical Robinson–Foulds (RF) metric[80] (calculated using scikit-bio's "compare_rfd" function) and the quartet score (calculated using ASTRAL) to quantify the topological concordance between a pair of trees. Furthermore, we used the "tip distance" (TT), calculated using scikit-bio's "compare_tip_distances" function, to measure the correlation of the phylogenetic distances among taxa in a pair of trees. It equals $(1 - r)/2$, where $r$ is the Pearson correlation efficient between the tip-to-tip distance (i.e., total length of branches connecting two tips) matrices of the two trees. Because the two trees might have different sets of taxa, we first truncated them using the "shear" function implemented in scikit-bio so that they both only contained the shared taxa. This enabled the subsequent computation of the three metrics.

For a set of multiple trees (species trees or gene trees), a matrix of the pairwise RF distance, quartet distance (1−quartet score) or tip distance was constructed, based on which subsequent statistical analyses were performed to assess the clustering pattern of trees, as stated below.

**Clustering analysis of multiple trees**. We used several statistical approaches to assess the clustering pattern of multiple trees based on the RF, tip or quartet distance matrices built as stated above:

1. Hierarchical clustering, using the "linkage" function implemented in SciPy 1.1.0[81].
2. mMDS, as detailed above.
3. Principal coordinate analysis (PCoA), performed using QIIME 2's "pcoa" command, and visualized using Emperor. This method aims to visualize the biggest variance in a few dimensions, as compared with mMDS as explained above.
4. Permutational multivariate analysis of variance (PERMANOVA)[82], performed using QIIME 2's "beta-group-significance" command, with 999 permutations (the default setting). This method evaluates the statistical significance of grouping of trees by a certain variable such as method, site sampling and taxon sampling.

**Cross-comparison of the ASTRAL and CONCAT trees**. The first challenge for this comparison was that the branch support values were estimated using completely different methods (local posterior probability vs. rapid bootstrap) and so are not directly comparable. We manipulated the trees so that they have the same overall resolution: First, we collapsed the low-supported branches in the CONCAT tree (by conserved sites), using the commonly accepted bootstrap threshold: 50. This left 9595 internal nodes. Then we performed branch collapsing to the ASTRAL tree, from the low end of the range of local posterior probability (lpp), until it reached 0.68057, also leaving 9595 internal nodes.

The second challenge was that large-scale trees are difficult to align and to display. We collapsed the two trees so that they have 50 paired clades with at least 50 descendants each. For each pair of clades, the descendants are identical. The remaining tips were pruned. This operation left 7764 taxa in each tree. The sizes of the 50 chosen clades are 155.3 ± 106.9 (mean and standard deviation).

A tanglegram of the resulting collapsed trees was reconstructed using Dendroscope 3.5.9[83]. In our case, the clades were fully aligned. The tanglegram was then rendered back-to-back without the need for displaying the connector lines.

**Calculation of the relative Archaea–Bacteria distance**. We calculated the phylogenetic distance (sum of branch lengths) between every pair of taxa in a tree using scikit-bio's "tip_tip_distances" function. The pairs were divided into three groups: A–A, A–B, and B–B (A and B are abbreviations for Archaea and Bacteria). Within each group, the mean distance was calculated. Then the overall relative A–B distance was calculated as: mean $(A–B)^2$/(mean $(A–A)$ × mean $(B–B)$). Note that due to HGT and other reasons, archaeal and bacterial taxa are rarely perfectly separated in individual gene trees. Therefore, the calculated distance should be interpreted as the average evolutionary distance between archaeal and bacterial genomes, instead of the distance between the two clades.

**Test for amino acid substitution saturation**. We followed the principle introduced by Jeffroy et al.[84] to test for the saturation. Specifically, we wanted to test whether the degree of saturation on inter-domain taxon pairs (Bacteria vs. Archaea) is larger than that on intra-domain pairs. For each domain, 100 taxa were randomly sampled for this analysis. We plotted the phylogenetic distance, i.e., the sum of branch lengths between two tips, as the $x$-axis, versus the Hamming distance of gap-free sites per each alignment between a pair of sequences, as the $y$-axis (Supplementary Fig. 19a–d). Because the three categories of taxon pairs have differential distribution on the $x$-axis, we further binned on the $x$-axis and performed comparison within each bin (Supplementary Fig. 19e, f).

**Phylogenetic analysis with latest genome availability**. We made several modifications to the main protocol to reduce the computational expense for this rapid test of the extended set of 10,762 (10,575 + 187) genomes: UPP was called in "insertion" mode to update the existing amino acid sequence alignments. In-house scripts were used to locate the same set of sites instead of performing de novo site sampling. Both ASTRAL and CONCAT methods were used to build species trees. For CONCAT, we used IQ-TREE in "fast" mode to build de novo species trees from concatenated alignments without using a predefined starting tree. For ASTRAL, we kept the same analysis parameters to build a species tree from the 381 gene trees, whereas the gene trees were built as follows to save computation while maintaining high quality:

First, we used the previous gene trees as topological constraints (-g) to incorporate the new taxa using RAxML. Then we used those trees as starting trees (-t) to perform de novo ML searches using RAxML. This way, we only did de novo ML search once instead of three as previously, but we argued that the generated gene trees would have comparable ML score as in the previous procedure. To test this hypothesis, we randomly selected ten genes to generate four trees each: (1) RAxML with FastTree tree as starting tree; (2) & (3) RAxML with random starting trees with two different random seeds; (4) RAxML tree generated using the described procedure. Note that the tree having highest likelihood score among (1), (2), and (3) defines the ML tree in the previous procedure. Our results showed that the gene trees generated by (4) have higher likelihood scores than the best of (1), (2), and (3) in six of ten of the tested genes. Besides, we use a $\chi^2$ test to show that the trees (4) have higher chance to be the best tree than (1), (2), and (3). In this test, the null hypothesis $H_0$ is that (4) has the same chance to be the best tree among the four trees. Applying the test on the ten selected genes, we rejected $H_0$ with $p$-value = 0.011.

**Divergence time estimation using maximum likelihood**. We used the maximum likelihood tool r8s 1.81[85] to estimate the divergence times based on the species trees. Specifically, we used the Langley–Fitch (LF) method[86], which assumes a universal molecular clock (substitution rate) for the entire tree, with the truncated-Newton (TN) method for optimizing the likelihoods of branch lengths[87]. A recent study showed that this method has comparable estimation accuracy when benchmarked against the more sophisticated Bayesian framework, but its computation is significantly faster[88], thus suitable for the size of our data set. Near-zero branches were collapsed to avoid numerical errors. Ten replicates with random initial conditions were performed for each setting. In each replicate, three restarts were executed after the initial optimization with a random perturbation factor of 5%. Replicates that failed to pass the gradient check were discarded. The divergence times estimated by the run with the highest likelihood score, and the mean and standard deviation of those by all successful runs were reported.

**Divergence time estimation using Bayesian inference**. We used the Bayesian tool BEAST 1.10.4[89] to estimate divergence times. Considering the computational expense, we randomly selected 5000 amino acid sites from the full-length alignment, and downsampled the original 10,575 taxa to 100. Taxon sampling was performed using the same RED-guided protocol (see above), but was manually modified afterwards to ensure sufficient sampling around the calibration point. Two alternative molecular clock models were used: the strict clock model, or the uncorrelated relaxed clock model with a lognormal distribution (UCLD)[90]. The species tree was modeled using a Yule process[91], with topology fixed to the ASTRAL tree. Logs of MCMC runs were examined using Tracer 1.7.1[92]. Burn-ins were set to be at least 10% of iterations, or higher depending on the manual observation of traces. Sufficient MCMC iterations were executed to ensure that the effective sample size (ESS) of the reported parameters was at least 150.

**Tree-based taxonomic curation and annotation**. We used the program tax2tree (commit 99f19be)[93] to curate the original NCBI taxonomy[94] assignment of genomes based on the phylogenetic trees and to annotate the internal nodes of the tree using most appropriate taxonomic labels. The same program was used to curate multiple databases, such as the classical Greengenes[93] and the recent GTDB[14]. The program took as input the species tree and the original NCBI taxonomy and inferred the most plausible taxonomic annotation at every node of the tree, as determined using an *F*-measure scoring system across candidate taxonomic terms. In scenarios where one term was estimated to be the best candidate for multiple, independent clades (i.e., para/polyphyly), a numeric suffix was appended to the term to indicate the grouping and order (from more descendants to less) of those clades. For example, Firmicutes_1 is the largest clade assigned to the paraphyletic phylum Firmicutes, followed by Firmicutes_2, Firmicutes_3, etc. Based on the decorated tree, correct taxonomic names were re-generated for unclassified and mis-annotated genomes. Taxonomic groups represented by only one genome in this work were back-filled post tax2tree annotation.

**Assessment of cladistic properties of taxon sets**. The cladistic property of a taxonomic group (or an arbitrarily defined taxon set) with reference to a species tree was evaluated using three methods:

1. The strict definition of "monophyly": when a clade contains all genomes assigned to a single taxonomic group and no other genomes, this taxonomic group is considered monophyletic. Further, we identified "relaxed" monophyletic groups compared to the aforementioned "strict" scenario. In the "relaxed" scenario, if a clade consists of genomes assigned to a taxonomic group, and genomes without assignments at the same taxonomic rank (i.e., unclassified), this taxonomic group is still considered monophyletic.

2. tax2tree's classification consistency score, representing the fraction of tips within that clade relative to the total number of tips in the tree which are of that taxon. Consistency = 1 is equivalent to strict monophyly.

3. The ASTRAL-computed quartet score of this taxonomic group, i.e., the fraction of quartets in the tree that supports this taxonomic group as monophyletic, i.e., separates this taxonomic group from the others.

4. An approach introduced in DiscoVista[95] which evaluates and visualizes the compatibility between a given taxon set and a tree with branch support values. It computes a "support" or "rejection" degree as follows: If the taxon set constitutes a monophyletic clade in the tree, it is supported; and the support degree (green) is the support value of the branch connecting the lowest common ancestor of the clade to its parent. On the other hand, if it is not a monophyletic group in the original tree, but after contracting branches with support values below a threshold, the monophyly can no longer be rejected due to polytomy, the lowest threshold is considered the rejection degree (with a negative sign) (magenta).

**Evaluation of GTDB taxonomic groups**. We downloaded GTDB[14] release: 86.1 from http://gtdb.ecogenomic.org/. The format of genome identifier in GTDB was matched to that of our work (e.g., GB_GCA_000123456.1 was translated into G000123456). Following the protocols described above, we evaluated the GTDB phylogeny and taxonomic units, and annotated our species trees using the GTDB taxonomy.

**Statistics**. Statistical analyses and plotting were performed using Python 3.6 and QIIME 2 release 2017.12. Specifically, PERMANOVA test was performed using QIIME 2's "beta-group-significance" command. Independent or paired two-sample *t* test was performed using scipy 1.1.0's "ttest_ind" and "ttest_rel" commands, respectively. Fisher's exact test was performed using scipy's "fisher_exact" function. Linear regressions were performed using scipy's "linregress" function. The *p*-value was computed using a two-sided Wald test, in which the null hypothesis is slope = 0. Gaussian kernel density estimations were performed using seaborn 0.9.0's "distplot" function. Hierarchical clustering was performed using scipy's "linkage" function. Quantile–quantile (Q–Q) plot was computed using scipy's "probplot" command. Redundancy analysis (RDA) was performed using vegan 2.4.4's "rda" and "ordiR2step" commands. Dimension reductions were performed using mMDS implemented in scikit-learn 0.19.2, or PCoA implemented in QIIME 2 (both detailed above). Pairwise distances based on *k*-mer signatures and on marker gene presence/absence were computed using the Jaccard index (see above). Branch supports in the phylogenetic trees were computed using rapid bootstrap implemented in RAxML 8.2.10, and ultrafast bootstrap implemented in IQ-TREE 1.6.1, and local posterior probability implemented in ASTRAL 5.12.6a (detailed above). Robinson–Foulds (RF) distance and "tip distance" were calculated using scikit-bio 0.5.2. Quartet scores were calculated using ASTRAL.

**Reporting summary**. Further information on research design is available in the Nature Research Reporting Summary linked to this article.

## Data availability

The data sets generated and analyzed during the current study are publicly available at GitHub (https://github.com/biocore/wol) and Zenodo (https://doi.org/10.5281/zenodo.3524546), under the BSD 3-Clause license. The source data underlying Figs. 1–6 and Supplementary Figs. 1–27 are provided as a Source Data file. All relevant data are available from the corresponding author.

## Code availability

The Python implementations of the prototype selection algorithms for genome subsampling are publicly available at GitHub (https://github.com/biocore/wol) and Zenodo (https://doi.org/10.5281/zenodo.3524546), under the BSD 3-Clause license. A copy of the code is provided in Supplementary Software.

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

## Acknowledgements

We are grateful to Victor Nizet, Eric Allen, Milton Saier, Jürgen Schulze, Toni Gabaldón Estevan, Benjamin Hillmann, Arturo Medrano-Soto, Madhusudan Gujral, Monika Kumaraswamy, Jeff Dereus, Anupriya Tripathi, Amnon Amir, Serene Jiang, Yoshiki Vázquez-Baeza, Thant H. Zaw for insightful discussions on this study and additional assistance. This study was supported in part by National Science Foundation grant 1565057 and the Alfred P. Sloan Foundation under grant number G-2017-9838 awarded to R.K. This work used the Comet supercomputer at the San Diego Supercomputer Center through allocation BIO150043, awarded to R.K., L.S. and S.M. through the Extreme Science and Engineering Discovery Environment (XSEDE). Both Comet and XSEDE are funded by the National Science Foundation. S.M., U.M., and M.R. were supported by the National Science Foundation grant III-1845967 to S.M. W.L. was supported by National Natural Science Foundation of China (91951205).

## Author contributions

Q.Z., R.K. and S.M. conceived the study; Q.Z. and U.M. led data analysis; U.M. and S.J. led algorithm development and implementation; W.P., U.M. and F.A. led high-performance computing; Q.Z. led the results interpretation and paper writing; Q.Z., G.A.A., J.B., Z.W., E.S., M.R., K.C., Y.Y. and S.M. contributed to algorithm development and implementation; U.M., S.J., J.G.S., P.B., D.M., S.H., N.S., J.J., S.P., C.H., S.M. and R.K. contributed to the result interpretation; F.A., T.K., J.T.M. and S.M. contributed to data analysis; Q.Z., F.A., E.K. and Z.Z.X. contributed to data collection; J.G.S., D.M., D.K., W.L., N.S., L.S., S.M. and R.K. contributed to study design; all coauthors contributed to the writing and discussion of the paper.

## Competing interests

The authors declare no competing interests.
