## [Peer Review File · Nature Communications]

Reviewers' comments:

Reviewer #5 (Remarks to the Author):

The manuscript presents fascinating evidence that the evolutionary distance between Bacteria and Archaea may be greatly overestimated due to an atypically large number of substitutions along this branch in ribosomal proteins compared to other near-universal markers.

This result, with some further analysis, could in my opinion form the basis of a manuscript fit for publication in Nat. Comms. It would be important to establish, however, if e.g. the results hold even if CPR sequences are not included?

Unfortunately, the second major result of the paper, involving placement and correspondingly the phylogenetic importance of CPR is more problematic.

My first and most important objection concerns the adequacy of the substitution models used. I agree with the previous rounds of review that placing CPR with confidence wr. other bacteria is a difficult phylogenetic problem, first and foremost because of the strong likelihood of LBA effects, which are expected to produce exactly the phylogeny the authors claim to find strong support for (i.e. CPR branching as a sister taxa to all other bacteria), as the long branch separating the two domains (even if it is shorter than suggested by ribosomal genes) can be expected to attract the CPR clade which contains long branches of its own. I also agree that currently the best, and practically the only, avenue to mitigate LBA is the use of sophisticated substitution models that are better able to capture the complexity of the sequence data. In particular, site-heterogeneous models, first and foremost among them the CAT+GTR model implemented in PhyloBayes has been demonstrated to be effective in overcoming LBA across a wide range of phylogenetic time-scales. This approach, however, is limited in the number of taxa it can analyze and a result several "short cuts" have been developed, with the currently most promising method for analyzing large concatenates being ML methods with empirical mixture models, in particular, IQTree with the C60 model. In my opinion, this would be the minimal model complexity required to place CPR in the context of Bacterial diversity. This is all the more so if Archaea are included as a putative outgroup.

A convincing demonstration of the basal position of CPR would involve recovering basal CPR under C60 and demonstrating that the branching order among major bacterial clades is not affected by the archaeal outgroup, e.g. by constructing a bacteria only concertante.

Using the PMSF model is also a possibility, but is significantly less convincing, due to the currently unexplored effect employing a guide tree, which in this case is likely to be distorted by LBA. Regardless even under the PMSF model it would be important to demonstrate that the branching order among major bacterial clades is not affected by the archaeal outgroup.

The second problem involves the use of ASTRAL. I find this deeply problematic for two reasons, first of all, the gene trees used as inputs to ASTRAL were produced using a pipeline that had to make serious compromises in the interests of tractability. This is demonstrated most clearly by the construction of the pipeline, which involves choosing the most likely among 3 ML trees based on alternative starting trees, which in itself, of course, strongly suggests that none of the trees is, in fact, the true ML tree, and offers no evidence that any of them are even close.

At the gene level, it would be equally, if not more important to use site-heterogeneous models (e.g. IQTree and C60). It would be equally important to avoid LBA prone starting trees! The two

parsimony starting trees are in this respect, and probably every other, worse than the FastTree one, but it would be ideal to start the ML search from random starting trees instead. Starting from several random starting trees a demonstration that the resulting putative ML trees are similar both in terms of likelihood and topology at higher taxonomic levels would be reassuring given the extent to which using ~10k taxa pushes the applicability of available methods.

Unfortunately, this is unlikely to be the case, as already suggested by the author's results: "Because of numerical instability, at least one of the RAxML runs failed for 39 of the 381 gene families. For those cases, IQ-TREE 1.6.1 75, an alternative and faster maximum likelihood program, was used instead to optimize branch lengths using the same model (G4). The tree with the highest likelihood score among the three runs was retained for downstream applications. In 161 gene families, this tree was from the run with the FastTree starting tree, while in the remaining gene families the best tree was from either one of the random seeds. "

Second, and more fundamentally, ASTRAL models gene tree incongruence resulting from ILS and only ILS, while at the scale of the ToL we have no reason to believe that ILS is a significant source incongruence, and at such phylogenetic scales, it is unlikely to be the most significant one, even if putative orthologs are considered. Simply pouring in a few hundred gene trees that, due to data size, are most probably poorly estimated, likely harbor bona fide incongruence not resulting from ILS, and as a result not modeled by ASTRAL, is not, in my opinion, a valid approach for producing a reference phylogeny.

In summary, reading the manuscript left me deeply skeptical that a reference phylogeny where the deepest relationships among Bacteria are resolved with confidence can be produced at the scale of ToL using ~10k taxa. The details of the methods used by the authors, in particular, the severe modeling compromises they had to make in the interests of tractability are such that the placement of CPR at the base of Bacteria is not convincingly established.

I would find a combination of fewer taxa and better models much more convincing.

I did, however, find the results showing that the branch between Bacteria and Archaea may be greatly overestimated to be very interesting. This is perhaps worth exploring in its own right.

Reviewer #5 (Remarks to the Author):

The manuscript presents fascinating evidence that the evolutionary distance between Bacteria and Archaea may be greatly overestimated due to an atypically large number of substitutions along this branch in ribosomal proteins compared to other near-universal makers.

This result, with some further analysis, could in my opinion form the basis of a manuscript fit for publication in Nat. Comms. It would be important to establish, however, if e.g. the results hold even if CPR sequences are not included?

We appreciate the reviewer's careful evaluation and valuable feedback on our manuscript. We are especially grateful that the reviewer found our evidence supporting the evolutionary proximity between Bacteria and Archaea "fascinating".

Following the reviewer's suggestion, we performed a new analysis, in which we removed CPR taxa from the dataset and rebuilt phylogenetic trees from the updated alignments ("*de novo*"). For comparison, we pruned the CPR clade from the main results ("*pruned*"). This new analysis is discussed in lines 197-202, Table S3 and Fig. S16. For the convenience of the reviewer, we pasted Fig. S16 below.

The result supports our main conclusion of the evolutionary proximity between domains Archaea and Bacteria. Specifically, the two metrics calculated using either "*de novo*" or "*pruned*" trees are close. In either group, the branch length and tip-to-tip distance between the two domains are remarkably shorter in trees based on the 381 global marker genes, using either the most conserved or randomly selected sites, as compared to trees based on the 30 ribosomal proteins.

Figure S16. Evolutionary distance between domains Archaea and Bacteria as revealed by phylogenetic trees without CPR taxa. Two metrics are assessed: **A.** Length of the branch connecting LCA of Archaea and LCA of Bacteria, normalized by the tree radius (calculated as the median of root-to-tip distances of all taxa). **B.** Relative Archaea-Bacteria distance, calculated as: $\text{mean}(A-B)^2 / (\text{mean}(A-A) \times \text{mean}(B-B))$, in which each distance is the sum of lengths of branches connecting one tip to another. Two sets of trees are compared: "pruned" are the same trees from the main results, with the CPR clade pruned; "de novo" are trees reconstructed from CPR-free sequence alignments. Each set contains six trees, built using either ASTRAL (A) or CONCAT (C), with either conserved or random site sampling from the 381 global markers, or with the 30 ribosomal proteins.

Unfortunately, the second major result of the paper, involving placement and correspondingly the phylogenetic importance of CPR is more problematic.

Following the suggestion of the reviewer and editor, we have completely removed this result from the manuscript. Now we only focus on the result of the proximity between Archaea and Bacteria.

Specifically, we deleted the entire section “*Archaea, CPR, and Eubacteria are three major clades*” and relevant discussion; removed Figs. 3C and S16 (PCoA of marker gene profiles), Fig. 3A-B’s inset plots and Table S3 (dimensions of the three major clades), Table S4 (placements of CPR in multiple trees); Table S5 (separation of gene profiles for the three groups), Fig. S24A-D (dimensions of the three clades by substitution model), as well as making necessary edits throughout the work to make this clear. In the only place where we still briefly addressed the placement of CPR in our trees, we added a statement: “*Considering the potential impact of long branch attraction, this placement will require further validation using more robust substitution models and controlled tests.*” (line 147). Finally, we changed the title of this manuscript into “*Phylogenomics of 10,575 genomes reveals evolutionary proximity between domains Bacteria and Archaea.*”

After these modifications, the reviewer’s comments below are now less relevant. However, we appreciate these insightful comments and provide detailed responses to them, including several additional analyses. These new results are only presented in this response letter, but not added to the manuscript. We will strive to elucidate the evolutionary status of CPR in more depth in a future manuscript, as the reviewer and the editor both suggested.

My first and most important objection concerns the adequacy of the substitution models used. I agree with the previous rounds of review that placing CPR with confidence wr. other bacteria is a difficult phylogenetic problem, first and foremost because of the strong likelihood of LBA effects, which are expected to produce exactly the phylogeny the authors claim to find strong support for (i.e. CPR branching as a sister taxa to all other bacteria), as the long branch separating the two domains (even if it is shorter than suggested by ribosomal genes) can be expected to attract the CPR clade which contains long branches of its own. I also agree that currently the best, and practically the only, avenue to mitigate LBA is the use of sophisticated substitution models that are better able to capture the complexity of the sequence data.

In particular, site-heterogeneous models, first and foremost among them the CAT+GTR model implemented in PhyloBayes has been demonstrated to be effective in overcoming LBA across a wide range of phylogenetic time-scales. This approach, however, is limited in the number of taxa it can analyze and a result several "short cuts" have been developed, with the currently most promising method for analyzing large concatenates being ML methods with empirical mixture models, in particular, IQTree with the C60 model. In my opinion, this would be the minimal model complexity required to place CPR in the context of Bacterial diversity. This is all the more so if Archaea are included as a putative outgroup.

A convincing demonstration of the basal position of CPR would involve recovering basal CPR under C60 and demonstrating that the branching order among major bacterial clades is not affected by the archaeal outgroup, e.g. by constructing a bacteria only concertante.

Using the PMSF model is also a possibility, but is significantly less convincing, due to the currently unexplored effect employing a guide tree, which in this case is likely to be distorted by LBA. Regardless

even under the PMSF model it would be important to demonstrate that the branching order among major bacterial clades is not affected by the archaeal outgroup.

We thank the reviewer for explaining the potential impact of LBA on the placement of CPR and the current state-of-the-art for tackling the LBA problem.

We agree that complex models are advantageous in ruling out LBA. However, as both reviewers noted, the elevated computational expense limits the data size, thus diverting us from the initial goal of comprehensive taxon and site sampling. Also, as we showed (and was appreciated by reviewer #1), reduced taxon sampling impacts recovered higher-level relationships (Text S2.4 and Fig. S12). Therefore we did not pursue this direction initially.

In the previous round of revision, we reported trees built using the PMSF method—which was recommended by reviewer #2—on 1,000 taxa. The results support our main conclusions (detailed in section “*Heterogeneity across sites*”). In choosing to use PMSF, we relied on the observation that the PMSF authors carefully investigated the effect of guide trees on the estimation, especially the resistance to LBA (Wang et al., 2018, *Systematic Biology*). They found that “*even when the incorrect guide tree was used*”, PMSF exhibited very small LBA bias, and was substantially more accurate than not using PMSF, and in some cases even more accurate “*than the mixture models from which they derive*.” They also found that “*with increasing taxon sampling, the estimation accuracies increase rapidly*.”

We also did analyze the data with more complex models, namely **IQ-TREE with C60** (not PMSF), as the current reviewer recommended. To do so we had to further downsample the data to 100 taxa, selected to maximize the representation of deep phylogeny (see “*Phylogeny-based downsampling of taxa*” in Methods), and 5,000 amino acid sites, selected randomly. This reduced dataset contains 12 Archaea, 16 CPR and 72 Eubacteria taxa. Per the reviewer’s suggestion, a separate analysis was performed with Archaea excluded from the dataset. Please find below the resulting trees, in which the basal placement of CPR is supported (UFBoot = 100 in both trees) (Fig. X1).

Figure X1. Phylogenetic trees reconstructed using the mixture profile model C60 on a reduced dataset. **A.** Tree of 100 taxa, including 12 Archaea, 16 CPR and 72 Eubacteria. **B.** Tree of the same CPR and Eubacteria taxa but excluding Archaea taxa. Branch support values are percentages of 1,000 ultrafast bootstrap replicates. Branches without labels are fully supported. The scale bar represents number of substitutions per site.

The second problem involves the use of ASTRAL. I find this deeply problematic for two reasons, first of all, the gene trees used as inputs to ASTRAL were produced using a pipeline that had to make serious compromises in the interests of tractability. This is demonstrated most clearly by the construction of the pipeline, which involves choosing the most likely among 3 ML trees based on alternative starting trees, which in itself, of course, strongly suggests that none of the trees is, in fact, the true ML tree, and offers no evidence that any of them are even close.

At the gene level, it would be equally, if not more important to use site-heterogeneous models (e.g. IQTree and C60). It would be equally important to avoid LBA prone starting trees! The two parsimony starting trees are in this respect, and probably every other, worse than the FastTree one, but it would be ideal to start the ML search from random starting trees instead. Starting from several random starting trees a demonstration that the resulting putative ML trees are similar both in terms of likelihood and topology at higher taxonomic levels would be reassuring given the extent to which using ~10k taxa pushes the applicability of available methods.

Unfortunately, this is unlikely to be the case, as already suggested by the author's results: "*Because of numerical instability, at least one of the RAxML runs failed for 39 of the 381 gene families. For those cases, IQ-TREE 1.6.1 75, an alternative and faster maximum likelihood program, was used instead to optimize branch lengths using the same model (G4). The tree with the highest likelihood score among the three runs was retained for downstream applications. In 161 gene families, this tree was from the run with the FastTree starting tree, while in the remaining gene families the best tree was from either one of the random seeds.*"

We thank the reviewer for discussing the potential impact of gene trees. In our original analyses, we used the standard approaches for generating starting trees in FastTree, RAxML, and IQ-TREE, which are used by most researchers. We went further and built three trees per gene, and selected the best tree by likelihood score for subsequent analysis. Also note that RAxML's default starting trees are randomized parsimony trees (stepwise addition), not maximum parsimony trees.

However, we performed an analysis using random starting trees as the reviewer suggested to test whether they result in substantially different results from our approach. Considering RAxML's computational expense, we randomly selected 10 marker genes, and reconstructed gene trees using RAxML. For each gene, we did 10 replicates using different seeds (so that RAxML starts from 10 random step-wise **parsimony** trees), and 10 replicates using different and completely **random** starting trees. We compared the Gamma likelihood scores of the 200 gene trees. We found that in 7 out of 10 genes, the mean likelihood of the "parsimony" group is higher than that of the "random" group (Fig. X2). This observation suggests that using parsimony starting trees is at least as effective as using random starting trees.

Figure X2. Comparison of likelihoods of RAxML gene trees inferred from parsimony starting trees (blue) or random starting trees (orange). Ten replicates using different seeds per group per gene were performed. Log likelihood values were computed under the Gamma model. To obtain comparable scales,

the minimum value of the 20 runs of each gene was subtracted from each value, then each value was divided by the maximum value of each gene.

We did not compare higher taxonomic levels of individual gene trees. This is because gene trees are highly discordant, as we demonstrated in section “*Heterogeneity among individual genes’ evolutionary histories*”, due to effects such as horizontal gene transfers as we and multiple reviewers have pointed out. For example, as we stated in our previous response to reviewer #2, Archaea and Bacteria are completely separated in only 19 out of 381 gene trees. This makes annotating gene trees with higher taxonomic groups very difficult. Therefore, in this manuscript, we based our discussion of deep phylogeny on species trees only.

Second, and more fundamentally, ASTRAL models gene tree incongruence resulting from ILS and only ILS, while at the scale of the ToL we have no reason to believe that ILS is a significant source of incongruence, and at such phylogenetic scales, it is unlikely to be the most significant one, even if putative orthologs are considered. Simply pouring in a few hundred gene trees that, due to data size, are most probably poorly estimated, likely harbor bona fide incongruence not resulting from ILS, and as a result not modeled by ASTRAL, is not, in my opinion, a valid approach for producing a reference phylogeny.

ASTRAL is a non-parametric method for summarizing gene trees based on minimizing the quartet distance from input gene trees to the output species tree. It is true that ASTRAL has proofs of consistency under ILS. However, it also has been proven consistent under stochastic models of HGT. Moreover, ASTRAL has been shown to work well under very high levels of HGT in simulations. See the paper below.

- Davidson, Ruth, Pranjal Vachaspati, Siavash Mirarab, and Tandy Warnow. “Phylogenomic Species Tree Estimation in the Presence of Incomplete Lineage Sorting and Horizontal Gene Transfer.” *BMC Genomics* 16, no. Suppl 10 (2015): S1. <https://doi.org/10.1186/1471-2164-16-S10-S1>.

The proof of consistency of ASTRAL under HGT (theorem 3 of paper above) follows from results of the following paper.

- Roch, Sebastien, and Sagi Snir. “Recovering the Treelike Trend of Evolution Despite Extensive Lateral Genetic Transfer: A Probabilistic Analysis.” *Journal of Computational Biology* 20, no. 2 (2013): 93–112. <https://doi.org/10.1089/cmb.2012.0234>.

The reviewer is right that ILS is less relevant to our case. However, they are mistaken that ASTRAL only is applicable when ILS is a concern. As the paper above shows, under random models of HGT, ASTRAL remains consistent and performs well. As the reviewer surely agrees HGT is a concern for our tree and this encourages the use of ASTRAL.

In summary, reading the manuscript left me deeply skeptical that a reference phylogeny where the deepest relationships among Bacteria are resolved with confidence can be produced at the scale of ToL using ~10k taxa. The details of the methods used by the authors, in particular, the severe modeling compromises they had to make in the interests of tractability are such that the placement of CPR at the

base of Bacteria is not convincingly established. I would find a combination of fewer taxa and better models much more convincing.

I did, however, find the results showing that the branch between Bacteria and Archaea may be greatly overestimated to be very interesting. This is perhaps worth exploring in its own right.

We thank the reviewer again for the careful review and insightful comments. As indicated above, we removed the discussion of CPR being basal to Bacteria and being a separate domain, and we refocused the manuscript on the Bacteria-Archaea proximity story. We greatly appreciate the reviewer's strong interest in it.

REVIEWERS' COMMENTS:

Reviewer #5 (Remarks to the Author):

The revised manuscript focusing on the distance evidence that the evolutionary distance between Bacteria and Archaea may be greatly overestimated due to an atypically large number of substitutions along this branch in ribosomal proteins compared to other near-universal markers is very interesting and thorough.

I do not have any major concerns and recommend publication.

Minor points:

Looking at the merged pdf with both OS X Preview and Adobe Reader Fig.1 has many missing characters and jumbled text in labels of the tree.

Referring to non-CPR bacteria as "Eubacteria" is confusing and I would advise it be revised to e.g. "non-CPR bacteria".

line 778 "biological rejection." is a term I am not sure is clear.

Response to Referee

Reviewer #5 (Remarks to the Author):

The revised manuscript focusing on the distance evidence that the evolutionary distance between Bacteria and Archaea may be greatly overestimated due to an atypically large number of substitutions along this branch in ribosomal proteins compared to other near-universal markers is very interesting and thorough.

I do not have any major concerns and recommend publication.

We thank the reviewer again for carefully reviewing the previous and current versions of our manuscript. The comments have greatly helped us to improve the quality of this work. We are glad to learn that the reviewer's major concerns have been resolved, and appreciate their recommendation for publication.

Minor points:

Looking at the merged pdf with both OS X Preview and Adobe Reader Fig.1 has many missing characters and jumbled text in labels of the tree.

We provide high-resolution PDF images as independent files in this final revision, which should be free of vector format issues.

Referring to non-CPR bacteria as "Eubacteria" is confusing and I would advise it be revised to e.g. "non-CPR bacteria".

Following the reviewer's suggestion, we have replaced "Eubacteria" with "non-CPR Bacteria" throughout the article.

line 778 "biological rejection." is a term I am not sure is clear.

We rephrased this statement as: *"But it should not be considered as a rejection to the monophyly of non-CPR Bacteria in the sense of evolutionary relationships."*